# Powerful Antibacterial Peptides from Egg Albumin Hydrolysates

**DOI:** 10.3390/antibiotics9120901

**Published:** 2020-12-13

**Authors:** Abdul-Raouf Al-Mohammadi, Ali Osman, Gamal Enan, Seham Abdel-Shafi, Mona El-Nemer, Mahmoud Sitohy, Mohamed A. Taha

**Affiliations:** 1Department of Science, King Khalid Military Academy, P.O. Box 22140, Riyadh 11495, Saudi Arabia; almohammadi26@hotmail.com; 2Biochemistry Department, Faculty of Agriculture, Zagazig University, Zagazig 44511, Egypt; aokhali@zu.edu.eg (A.O.); mzsitohy@zu.edu.eg (M.S.); 3Botany and Microbiology Department, Faculty of Science, Zagazig University, Zagazig 44519, Egypt; mohamednoor220@yahoo.com (M.E.-N.); dr.taha_virus@yahoo.com (M.A.T.)

**Keywords:** egg albumin, pepsin, antibacterial activity, pathogenic bacteria, hydrolysis, acute toxicity

## Abstract

Native egg albumin (NEA) was isolated from hen eggs and hydrolyzed by pepsin to produce hydrolyzed egg albumin (HEA). HEA was chemically characterized and screened for its antibacterial activity against 10 pathogenic bacteria (6 Gram (+) and 4 Gram (−)). The SDS-PAGE pattern of NEA showed molecular weights of hen egg albumin subunits ranging from 30 to 180 kDa. The highest intensive bands appeared at a molecular mass of about 50 and 97 kDa. Ultra-performance liquid chromatography (UPLC) of the peptic HEA revealed 44 peptides, 17 of them were dipeptides, and the other 27 fractions corresponded to bigger peptides (3–9 amino acids). The dipeptides and big peptides represented 26% and 74% of the total hydrolysate, respectively. The MIC of HEA was about 100 μg/L for *Listeria monocytogenes*, *Bacillus cereus*, *Staphylococcus aureus*, *Salmonella typhimurium*, *Streptococcus pyogenes*, and *Klebsiella oxytoca* and 150 μg/L for *Pseudomonas aeruginosa*, *Bacillus subtilis*, and *Listeria ivanovii* and 200 μg/L for *Escherichia coli. L. monocytogenes* was the most sensitive organism to HEA. Mixtures of HEA with antibiotics showed more significant antibacterial activity than individually using them. Transmission electron microscopy (TEM) revealed various signs of cellular deformation in the protein-treated bacteria. HEA may electrostatically and hydrophobically interact with the cell wall and cell membrane of the susceptible bacteria, engendering large pores and pore channels leading to cell wall and cell membrane disintegration. Higher cell permeability may, thus, occur, leading to cell emptiness, lysis, and finally death. Alternatively, no toxicity signs appeared when HEA was administrated to Wistar Albino rats as one single dose (2000, 5000 mg/kg body weight) or repeated daily dose (500 and 2500 mg/kg body weight/day) for 28 days to disclose the possible toxicity hazards. HEA did not produce any death.

## 1. Introduction

Natural antimicrobials from food sources are welcomed as food preservatives as they are not hazardous and have no side effects like those caused by synthetic ones [1,2,3,4,5,6]. These natural antimicrobials are quite promising for food bio-preservation to reduce the need for antibiotics, control of microbial spoilage process, and could kill the resistant variants of bacteria that can survive in foods and limit the occurrence of new food-borne disease outbreaks caused by pathogenic bacteria [7,8,9,10,11,12]. Natural extracts are compounds with a wide structural diversity, representing an important source of new chemical compounds with a possibly significant antibacterial action against food-borne pathogens with powerful action against multidrug-resistant bacteria [13,14,15,16,17,18,19,20,21]. The possible negative impact of such chemicals on the environment and human health may preclude synthetic agents’ use [19]. Therefore, novel antimicrobial agents from natural sources are highly required [4,15,17,20]. Many strategies have been suggested to improve proteins’ antimicrobial activities, including chemical modification, e.g., such as esterification [22,23,24,25,26]. Food proteins have long been recognized as important source of bioactive peptides. In particular, peptides derived from protein-based foods have been shown to possess significant antimicrobial capacities. These peptides from either plant or animal sources can be generated through enzymatic hydrolysis, acid or alkali treatment, or fermentation [26]. Bioactive peptides can be obtained from different protein sources, including milk [27] and legume proteins [28,29]. Peptides from various protein resources have been reported to have multiple functions, including antioxidative ability, angiotensin I converting enzyme (ACE) inhibitor properties [30], hepatoprotective activity [31], and antibacterial activity [6,19,20,32,33,34,35,36]. 

Eggs contain various proteins exhibiting antimicrobial activities, including lysozyme and ovotransferrin, which are commonly used as natural food antimicrobial agents against a wide spectrum of bacteria [37]. Natural antimicrobial peptides exist in different compartments (egg shell, albumen, and vitelline membranes) of eggs. Most of these peptides are cationic or amphipathic, but there are also hydrophobic α-helical peptides that possess antimicrobial activity [38]. Although a wide variety of peptides with different chemical structures and peptide conformations have been reported to exhibit antimicrobial activity, these antimicrobial activities’ mechanism is less well understood in most cases. In fact, several observations suggest that natural peptides can alter cytoplasmic membrane permeability, inhibit cell-wall synthesis, inhibit nucleic-acid synthesis, inhibit protein synthesis and in turn, cause bacterial cell death [39]. Therefore, antimicrobial peptides (AMPs) constitute a promising alternative as therapeutic agents against various pathogenic microbes [40]. More than 60 peptide drugs have reached the market for patients’ benefit, and approximately 140 peptide therapeutics are currently being evaluated in clinical trials [41]. In a previous study [37], the methylated egg white protein inhibited many bacterial pathogens, simulating antibiotics’ inhibitory activity. In an endeavor to develop such work, the present study aimed to determine the antibacterial activity and mode of action of protein hydrolysates produced from egg albumin by enzymatic digestion using pepsin. The toxicity of both native egg albumin (NEA) and hydrolyzed egg albumin (HEA) was studied using Wistar albino rats.

## 2. Results

### 2.1. Chemical Characterization of HEA

The SDS-PAGE patterns of HEA are shown in Figure 1 and Table 1. The numbers 1–5 refer to the protein fractions of our tested protein (egg albumin) from up to down successively. In Figure 1, the MW200, 106, 97, 66, 45, 30 refer to molecular weight standards in the first lane (St) in Figure 1. In Table 1 the MW 180, 144, 97, 50, 30, refer to the subunits of egg albumin as also shown in the second lane of Figure 1 (1). These molecular weights were deduced using the known standard MW in lane (1) It is evident that the molecular weight of NEA subunits ranged between 30 and 180 kDa. The more intensive bands were presented at molecular masses of about 50 and 97 kDa. The hydrolysate’s electrophoretic pattern indicated that the peptides formed due to hydrolysis were less than 50 kDa. Urea-PAGE patterns of HEA compared to NEA are presented in Figure 1. HEA (lane 2) migrated faster than NEA (lane 1) from anode to cathode due to increased positive charge on the hydrolyzed proteins. Native-PAGE and SDS-PAGE of native and modified proteins represented exchanges in molecular masses between native and modified proteins.

Since using chemical agents in the food and pharmaceutical industries can leave residual organic solvents or toxic chemicals in the products, enzymatic hydrolysis is widely preferred. The hydrolysis of egg albumin with pepsin was monitored for 240 min as recorded in Figure 2. The hydrolysates obtained after 60, 120, 180, and 240 min, exhibited about10%, 18%, 25%, and 3% degree of hydrolysis, respectively.

### 2.2. Ultra-Performance Liquid Chromatography (UPLC) of the Peptic Hydrolysate of Egg Albumin

Peptide profile of the peptic hydrolysates of egg albumin was obtained from Ultra performance liquid chromatography (UPLC) analysis (Acquity H Class, Waters) combined with quadrupole-time-of-flight mass spectrometry (Synap G2, Waters), revealed the presence of 44 peptides (Figure 3). The molecular weight of 17 fractions was in the range of dipeptides, representing about 26% of the total peptide fractions based on the corresponding fractions’ total areas. The other 27 peptides’ molecular weights corresponded to peptides ranging from tri-peptides to octa- or nona-peptides, forming about 74% of the total peptides, as concluded from the total areas of the corresponding peaks. The higher relative quantity of the bigger sized peptides (3–9 amino acids) may indicate their higher participation in the hydrolysates’ biological activity. The selected peptides summarized in Table included in Figure 3, were chosen as the major constituting peptides that are expected to produce the highest biological impact.

### 2.3. Antibacterial Activity of HEA and NEA against Gram (+) and Gram (−) Bacteria

HEA was prepared via enzymatic hydrolysis of egg albumin using pepsin at optimal conditions (pH 2 at 37 °C) to potentially produce cationic antimicrobial peptides with antibacterial effect. Using different HEA and NEA concentrations, Table 2 revealed that 250 μg/mL of HEA was the most inhibitory concentration against all the tested bacteria. Disc diffusion assay indicated concentration-dependent inhibition zone diameters for HEA. The MIC of HEA was 100 μg/mL against *L. monocytogenes*, *B. cereus*, *S. aureus*, *S. typhimurium*, *S. pyogenes*, and *K. oxytoca*, 150 μg/mL against *P. aeruginosa*, *B. subtilis*, and *L. ivanovii*, and 200 μg/mL against *E. coli*. The data in Figure 4 represent the 24 h growth curves of 10 pathogenic bacteria subjected to one value of MIC of HEA showing general growth inhibition. Among the Gram (+) bacteria, *B. subtilis* and *B. cereus* were the most inhibited organisms by HEA. While in the case of Gram (−) bacteria, *K. oxytoca* showed nearly complete growth inhibition during the 24 h period of incubation. However, there is a moderate effect of NEA on the 10 tested bacteria compared with the control. Therefore, HEA was the most inhibitor compared to NEA and control bacteria (Table 2 and Figure 4).

### 2.4. Antibacterial Activity of Antibiotics and HEA-Antibiotic Combinations by Disc Diffusion Assay

An antibiotic sensitivity test was carried out for all the tested pathogenic bacteria using five antibiotics listed in Table 3 and Figure 4. There was variability in the sensitivity of the tested pathogenic bacteria. The results given in Table 3 and Table 4 showed the MICs of antibiotics, HEA, and combinations of them. MICs values (μg/mL) were 7.5, 6, 4, 4, 2, 2, 1, 1, 1, 0.016 for the antibiotics chloramphenicol, gentamycin, ciprofloxacin, vancomycin, chloramphenicol, ciprofloxacin, vancomycin, ciprofloxacin, gentamycin, penicillin as determined against *L. ivanovii*, *K. oxytoca*, *S. typhimurium*, *B. cereus*, *B. subtilis*, *L. monocytogenes*, *S. aureus*, *P. aeruginosa*, *E. coli*, *S. pyogenes,* respectively. MICs of HEA were in the range 100–200 μg/mL depending on the bacteria. Mixtures of modified proteins at equal MIC of HEA and antibiotics showed greater antibacterial activity than any single treatment of either of them, referring to synergistic action. This effect was most evidently seen in the case of *L. monocytogenes* with the biggest inhibition zone (36 ± 2.31 mm) followed by *S. aureus* and *S. pyogenes* with the second biggest inhibition zones 34 ± 1.16 and 34 ± 0.58 μg/mL, respectively.

### 2.5. TEM Analysis

The presence of HEA (100 μg/mL) in BHI broth media containing *S. aureus* (Gram (+) bacterium), OD_600_ = 0.5 at the time of application produced TEM images showing reduced relative content of intact cells after 4 h of incubation at 37 °C (Figure 5). TEM images clarified that bacterial cells, escaping death, were subjected to different manifestations of deformation. Similar signs of effects on *P. aeruginosa* (Gram (−) bacterium), including cell shrinkage, cell membrane wrinkles, pore formation, and emptiness of live cellular material were observed in the HEA (150 μg/mL) treated cells. TEM results indicated that the cationic proteins’ action is rather general and more targeting the cell wall and cell membrane.

### 2.6. Investigation of Toxicity of HEA in Wistar Albino Rats

#### 2.6.1. Acute Oral Toxicity

No mortalities were recorded after oral administration of the native egg albumin (NEA) or its hydrolyzed forms (HEA), administered at different doses in the range of 2000–5000 mg protein/kg body weight/day. LD_50_ of HEA was higher than 5000 mg/kg body weight/day. Substances with an LD_50_ between 5000 and 15,000 mg/kg body weight/day are regarded non toxic.

No signs of toxicity, e.g., abnormal breathing and impaired movements, were observed in any rat group. Over the next 14 days, no adverse effects of these treatments were produced. The changes in body weight did not differ significantly between rat groups. They were in the same range 16–18g/rat in each rat group. The non observed adverse effect level is very high, being probably greater than 5000 mg/kg body weight/day Table 4.

#### 2.6.2. Subacute Toxicity

The data presented in Table 4 show relatively insignificant (*p* < 0.05) changes in the body weights of Wistar Albino rats receiving 5 days per week of low and high doses (500 and 2500 mg/kg body weight/day) of HEA-treated rats. Weight gain was in the same range 80–90 and 52–61g/rat for male and female, respectively. At the end of the experimental period, the organ weight of all rats was nearly in the same range without significant differences (*p* < 0.05, Table 5).

#### 2.6.3. Hematological Parameters

The changes in hematological parameters (WBS, RBC, hemoglobin, hematocrit, MCV, MCH, MCHC, platelets, and lymphocytes) were either within the normal range, insignificant and not-dose dependent in rats receiving two levels of HEA for 28 days (Table 6).

#### 2.6.4. Biochemical Indicators

The treatment with HEA for 28 days (500 and 2500 mg/kg body weight) did not change the biochemical profile of male and female rats, as shown in Table 7.

#### 2.6.5. Histo-Pathological Examination

The rat organs were histologically examined at the end of the experimental exposure of rats to the tested substances for any abnormalities. The rat organs were histologically examined at the end of the experimental exposure of rats to the tested substances for any abnormalities. The obtained results are shown in Figure 6 and Figure 7. The macroscopic analysis of the treated animals’ target organs did not show any histological alterations in all organs examined.

## 3. Discussion

The occurrence and spread of antibiotic resistant bacteria are pressing public health problems worldwide [13,34,42]. Many bacteria have become resistant against many antimicrobial agents. The resistance rates are higher in developing countries [43]. The development of new prophylactic and therapeutic procedures is urgently required to meet the challenges imposed by the emergence of bacterial resistance [44,45,46]. Cationic Antimicrobial Peptides (CAMPs) are promising new antibacterial agents due to their killing mechanism via interaction with bacterial cell walls and membranes [20,47].

Imparting cationic character on native proteins can be achieved through chemical modification, e.g., esterification which blocks free carboxyl groups, elevating the net positive charge on the modified proteins [48,49] or through the enzymatic hydrolysis [50] which potentially affects the molecular size and hydrophobicity, as well as the polar and ionizable groups of protein hydrolysates. This modification produces potent bactericidal peptides that may be classified as effective antimicrobials [51,52].

In line with this trend, HEA was obtained by enzymatic hydrolysis of egg albumin using pepsin at its optimal conditions (pH 2 at 37 °C) to liberate bioactive peptides, a relatively short peptide residue length (e.g., 2–20 amino acids) possessing hydrophobic amino acid residues in addition to proline, lysine, or arginine groups and bioactive peptides. These peptides showed to be an effective antimicrobial [52]. The large number of the peptides (44 fractions) released by peptic hydrolysis are apparently due to the broad specificity of pepsin and the high susceptibility of proteins to this enzyme [53]. The majority of the peptides which had relatively high molecular weight (3–9 amino acids) agrees with the fact that pepsin attacks the protein molecules gradually from the C- and N-terminals [53]. The obtained hydrolysate style, showing increasing and steady rate of hydrolysis with time during 4 h, reflects the nature of pepsin specificity which tends to attack protein molecules gradually from the C- and N-terminals [54,55], targeting preferentially linkages with aromatic or carboxylic L-amino acids [56]. The higher relative quantity of the bigger sized peptides may indicate their higher participation in the biological activity of the hydrolysates; protein hydrolysis may give rise to basic peptides which can directly attack the cell walls and cell membranes of the pathogenic bacteria [57,58].

The higher relative quantity of the bigger sized peptides (3–9 amino acids) may indicate their higher participation in the biological activity of the hydrolysates, i.e., peptides of more than three amino acids may be essentially more responsible for the antibacterial activity since they represent more than 74% of the peptide mixture. However, the data available from the analysis cannot give information about the specific peptide that was most responsible for this activity.

The observed molecular mass range of NEA subunits between 50 and 180 kDa agrees with its ability to dissociate into more active subunits under different ionic strength and pH [59,60,61,62]. The electrophoretic patterns of HEA confirmed the transformation of NEA into lower molecular weight peptides. The faster migration from anode to cathode, in case of the hydrolyzed protein, than the native one refers to bigger positive charges. This may imply the liberation of some cationic peptides in the course of hydrolysis in accordance with previous studies [63,64]. HEA exhibited a distinctive inhibition against the pathogenic bacteria, while NEA showed only scant inhibition corroborating previous results [65,66,67,68].

The higher antimicrobial activity of HEA than NEA could be due to the more hydrophobicity and cationic nature of HEA [26,37,69,70,71,72]; initiating the electrostatic interaction between the positive charges of the antimicrobial agent and the negatives charges on the bacterial cell walls or membranes. The hydrophobic aggregation between the similar regions of the two reactants can further stabilize this complex. Oscillating random Brownian motion of these aggregations [71] may produce large-sized pores channels and disintegrate the cell wall and cell membranes, engendering higher cell permeability, cell emptiness, and death. The mechanism of antimicrobial peptides is based on their amino acid composition, amphipathicity, and cationic charges allowing their attachment to cell membrane bilayers to form pores by ‘barrel-stave’, ‘carpet’, or ‘toroidal-pore’ mechanisms [37,39]. 

Since HEA shows broad-spectrum antimicrobial activity, it can promote the activity of antibiotics against all tested bacteria. This was realized in the tested HEA-antibiotic combinations which showed greater antibacterial action than the single antibiotics, especially against *L. monocytogenes*, *B. cereus*, and *P. aeruginosa*. This synergism between antibiotics and HEA may be due to accentuating the hydrogen bonding and hydrophobic–hydrophobic interactions with targeted microbes [73] or through using different mechanisms of bacteria inhibition.

This Gram (−) bacteria (especially *P. aeruoginosa*) is known for being notorious for its ability to survive in the environment, particularly in moist conditions. It may contaminate medicines, surgical equipment, clothing, and dressing with the ability to cause serious infections in immune compromised patients [74]. Therefore, it was used in this study for TEM studies. Additionally, *S. aureus* bacterium was selected from the Gram (+) group as it was more sensitive and because of its public health concerns. *S. aureus* is a pathogen of skin abscesses, pharyngitis, sinusitis, meningitis, pneumonia, endocarditis, osteomyelitis, toxic shock syndrome, sepsis, and wound infections following surgery. Moreover, it responsible for food poisoning illness as it is capable of producing several virulence factors such as enterotoxins, adhesins, hemolysins, invasins, superantigens, and surface factors that inhibit its phagocytic engulfment [73,75,76].

The signs of the irregular wrinkled outer surface, fragmentation, adhesion, and aggregation of damaged cells or cellular debris of HEA-treated bacterial cells, manifested by TEM examination confirm the antimicrobial action and follow previously published works [23,52,77].

The potential toxicity of NEA and HEA was assessed by determining acute oral toxicity when administrated to Wistar Albino rats. Mortality absence after a single administration of up to 5000 mg/kg body weight/day of the two tested substances indicates their safety and their possible biological utilization. The absence of significant body weight changes or any abnormality signs during 14 days after the single high dose administration of HEA (5000 mg/kg body weight/day) indicates its harmlessness. Since no deaths emerged in response to the used range of doses, the LD_50_ of HEA could not be calculated, but it may be assumedly more than 5000 mg/kg body weight/day. Therefore, HEA may be regarded nontoxic since its LD_50_lies in the range 5000–15,000 mg/kg body weight/day according to [78].

Extending the administration of two HEA doses (500 and 2500 mg/kg body weight/day) for 28 days did not either produce evident signs of toxicity or pathogenicity on the body or organ weights, indicating the absence of any specific organ toxicity. Changes in body or organ weights are taken as indicators of toxicity [79] while normal body weight changes are indicators of food safety and lack of toxicity [80]. The rats conserved normal healthy status in spite of repeated force-feeding with high doses. The obtained results revealed slight reductions in these two parameters and showed absence of any negative effects of the two studied substances on the renal function during repeated dose administration for 28 days. The rats conserved normal healthy status despite repeated force-feeding with high drug doses. The obtained results revealed the absence of any negative effects on the renal or hepatic function as well as the histological status during repeated-dose administration for 28 days in agreement with a previous study [81]. Therefore, our data demonstrate the importance of hydrolyzed egg albumin in controlling resistant bacteria. Further work will be necessary to study pathogenic bacteria’s inhibition by HEA in processed foods based on egg albumin.

## 4. Materials and Methods

### 4.1. Collection of Pathogenic Bacteria

*L. monocytogenes* LMG10470, *L. ivanovii* LMG11388, *B. subtilis* LMG7135, *B. cereus* ATCC14579, *S. aureus* DSM1104, *S. pyogenes* ATCC12384, *K. oxytoca* LMG3055, *P. aeruginosa* LMG8029, *E. coli* LMG8223, and *S. typhimurium* LMG10395 were used. All the bacteria were stored as stock cultures in glass beads at −20 °C and subcultured propagated in brain heart infusion broth (BHIB, Oxoid), before use. Slope cultures were grown on (BHIB, Oxoid, Columbia, MD, USA) as described previously by Abdel-Shafi S. et al., 2019a [34] and preserved stored at 4 °C during experimentation.

### 4.2. Egg Albumin Preparation

Fresh hen eggs were obtained from the Farm of Faculty of Agriculture, Zagazig University, Zagazig, Egypt. They were hand broken and the whites were separated from the whole eggs. The egg white was gently subjected to a magnetic stirrer for 30 min to reduce the viscosity. They were then lyophilized (Thermo-electron Corporation—Heto power dry LL 300 Freeze dryer). The lyophilized protein product was designated native egg albumin (NEA) according to Abdel-Shafi S. et al., 2016 [37].

### 4.3. Enzymatic Hydrolysis of Egg Albumin

Enzymatic hydrolysis of egg albumin was performed using pepsin at optimal conditions (pH 2 at 37 °C). The protein substrate was dissolved in the relevant buffer (0.1 M Glycine–HCl pH 2) at 10 mg/mL and the enzyme was added at 1:2 (*w/w*) enzyme-to-substrate ratio. After 4 h of reaction, the sample was heated in a boiling water bath for 10 min to inactivate the enzyme in accordance to Otte J. et al., 2015 [82]. The hydrolysates were centrifuged at 4000× *g* for 15 min and the supernatant was lyophilized and stored at −20 °C. 

#### Characterization of Egg Albumin Hydrolysates

The degree of hydrolysis (DH) (the percentage of peptide bonds cleaved), was calculated at different periods (0, 1, 2, 3, and 4 h) by determining free amino groups through reaction with L-Leucine, 2,4,6-trinitrobenzenesulphonic acid (TNBS) following Adler-Nissen J., 1986 [83].

SDS–PAGE of both NEA and HEA was carried out as described previously by Sitohy M. and Osman A. 2010 [84] in 3% and 17% acrylamide for the stacking and resolving gels, respectively, as described previously by Evans R.W and Williams J., 1980 [85]. 

Native egg albumin hydrolyzed with pepsin for 4 h was analyzed by Urea-PAGE in 3% stacking and 10% resolving gels according to CLSI, 2008 and Standards N.C.F.C.L. et al., 1999 [86,87].

Ultra performance liquid chromatography (UPLC) was conducted in the Center of Drug Discovery Research and Development, Faculty of Pharmacy, Ain-Shams University, Egypt, using (Acquity H Class, Waters) coupled with quadrupole-time-of-flight mass spectrometry (Synap G2, Waters). An aliquot of 10 µL of the final peptide solution was injected into the chromatograph and peptides were separated on an ACQUITY UPLC—BEH C18 1.7 µm—2.1 × 50 mm. The mobile phase combined deionized water with 0.1% formic acid as solvent A, while acetonitrile with 0.1% formic acid represented solvent B. The initial conditions were 95% A and 5% B to establish a 95% (*v/v*) linear gradient of B at 15 min. The running specifications were 25 min total run time, 5 min post-delay time, and 0.2 mL min^−1^ flow rate. A high-resolution mass spectrometric analysis was carried out in the positive electro spray ionization (ESI+). High-purity nitrogen was used for dissolution (800 L h^−1^) and cone (25 L h^−1^). Spectra were recorded over the mass/charge (*m*/*z*) range of 100–1000. 

### 4.4. Antibacterial Activity Estimation

#### 4.4.1. Disc Diffusion Assay

The antibacterial activity of both NEA and HEA were tested against the experimental pathogenic bacteria by the Kirby-Bauer disc diffusion method as described previously by Schafer E.W. et al., 1972 [88]. The bacterial suspensions were spread over the surface of nutrient agar plates, then sterilized paper discs of 6mm in diameter were soaked in either diluted NEA solutions (0, 10, 25, 50, 100, and 250 μg/mL) or HEA solutions (0, 50,100,150, 200, and 250 μg/mL). They were then laid onto the surface of nutrient agar media seeded with the pathogenic bacteria at appropriate distances, separating them from each other. The nutrient agar plates were incubated at 37 °C for 24 h before recording the results in accordance to OECD, 2001 [89].

#### 4.4.2. Minimum Inhibitory Concentration (MIC)

MIC of HEA and five antibiotics, ciprofloxacin (10 μg), gentamycin (10 μg), penicillin G (10 μg), chloramphenicol (30 μg), andvancomycin (30 μg) was evaluated using standard inoculums of about 2 × 10^5^ CFU/mL following Abdel-Shafi S. et al., 2019b [35]. Serial dilutions of the test compounds, previously dissolved in sterilized distilled water, were prepared to final concentrations to each tube (containing 10 mL BHI broth), 100 μL of the inoculum was added and incubated for 24 h at 37 °C. At the end of incubation time, MIC was visually identified as the lowest concentration of the test compound, inhibiting the visible growth as confirmed by measuring the optical density at 600 nm. Sterilized distilled water was used as a negative control.

#### 4.4.3. Antibacterial Activity of Antibiotics and Combinations of Both HEA and Antibiotics

Ready antibiotic discs (μg/disc) of ciprofloxacin (10 μg), gentamycin (10 μg), penicillin G (10 μg), chloramphenicol (30 μg), and vancomycin (30 μg) were laid onto the surface of nutrient agar media, seeded previously with all the tested bacteria, leaving appropriate distances separating them. The nutrient agar plates were incubated at 37 °C for 24 h, and the diameters of the inhibition zones (mm) were measured. Results of antibiotic sensitivity were taken according to OECD, 2001 and Osman A. et al., 2019 [89,90]. In another experiment, a disc of the more effective antibiotic against each bacterium was selected and was soaked in MIC μg/mL of HEA and placed onto surface of nutrient agar seeded with the indicator bacterium. After 48h of incubation at 37 °C, results were taken as mentioned above.

#### 4.4.4. Quantitative Inhibition of Pathogenic Bacteria by HEA

A number of test tubes, containing 10 mL of BHIB (Oxoid, United Kingdom), were inoculated with 100 µL of bacterial suspension (2 × 10^5^ CFU/mL) at the log phase and combined with 100 μg/mL HEA for *L. monocytogenes*, *B. cerueus, S. aureus*, *S. typhimurium, S. pyogenes,* and *K. oxytoca* and 150 μg/mL HEA for *P. aeruginosa*, *B. subtilis,* and *L. ivanovii* and 200 μg/mL HEA for *E. coli*. Control test tubes contained BHIB (Oxoid) using uniquely the bacteria without treatment. Samples and controls were incubated at 37 °C before determining the growth after 0.0, 6, 12, 18, and 24 h of incubations by measuring the turbidity at OD_600_ using a spectrophotometer (ENWAY—England 6405 UV/VIS, England).

### 4.5. Transmission Electron Microscopy Examination 

Morphological and ultrastructural changes of bacterial cells of *S. aureus* and *P. aeruginosa* were examined using transmission electron microscopy (TEM) (JEOL-TME-2100F, Japan) as described previously by Standards N.C.F.C.L. et al., 1999 [87]. Prior to TEM imaging, each bacterial strain was grown in tryptone glucose yeast extract broth supplemented with or without 100 μg/mL of HEA and incubated at 35 °C for 24 h.

### 4.6. Investigation of Toxicity for HEA in Wistar Albino Rats

Zagazig University Animal Care Board approved the design of the toxicological animal experiment (Approval number: ZU-IACUC/1/F/134/2020). Animals used and the design of the experiment were carried out following Osman A. et al., 2019 [90]. Healthy male and female white albino rats (*Rattus norvegicus*. Bork), Wistar strain (135 ± 10 g, bodyweight for female and 155 ± 15 g, bodyweight for male) were obtained from the Organization of Biological Products and Vaccine (Helwan farm, Cairo, Egypt) and housed in plastic cages in groups of five animals/cage.

The experimental animals were allowed to acclimatize under the laboratory conditions (temperature of 28 ± 2 °C; relative humidity 50–70% and normal light/dark cycle) for 2 weeks prior to the experiment. They were provided with balanced pelleted diet (23% protein) and tap water ad libitum throughout the adaptation and experimental period. The experimental design included two phases; the first phase was destined to assess the acute oral medium lethal doses (LD_50_) while the second one was to evaluate HEA subacute toxicity.

#### 4.6.1. Acute Oral Toxicity

Fifteen male Wistar Albino rats (160 ± 10 g, body weight) were divided into three groups (five rats each). The first group received 2 mL distilled water free from any external treatment (control). The groups 2 and 3 received one acute dose of HEA of 2000 and 5000 mg/kg body weight, respectively. Each group received a stomach tube force-feeding of different specified doses dissolved in distilled water (2 mL). All rats were kept under observation for 24 h and symptoms of toxicity or mortality were recorded. Animals that were still alive were observed daily for their behavior and body weight changes for 14 days. The acute oral LD_50_ values were calculated accordance to Osman A. et al., 2019 and Guidelines for Hematoxylin and Eosin Staining, 2001 [90,91].

#### 4.6.2. Subacute Toxicity 

Fifteen male Wistar Albino rats (160 ± 10 g, body weight) were divided into three groups (five rats each). The first group received 2 mL distilled water free from any external treatment (control). Groups 2 and 3 received one acute dose of HEA of 2000 and 5000 mg/kg body weight. Each group received a stomach tube force-feeding of different specified doses dissolved in distilled water (2 mL). All rats were kept under observation for 24 h and symptoms of toxicity or mortality were recorded. Animals that were still alive were observed daily for their behavior and body weight changes for 14 days. The acute oral LD_50_ values were calculated. Fifteen male rats were divided into three groups of five rats each. Each group was intubated orally by force-feeding using a stomach tube with different HEA doses five days/week for 28 days. The groups received the following doses: Group 1: the same amount of distilled water without any treatment and served as control. Groups 2 and 3: HEA at doses of 500 and 2500 mg/kg body (weight/day), respectively. The same experiment was applied to 15female rats. All animals, including control, were supplied with a balanced pelleted diet (23% protein) and tap water ad libitum during the experimental period. Daily body weight of rats was recorded, and the weight gains were calculated during the testing period as follows;
Body weight gain (g)=Final body weight (g)−Intial body weight (g)

At the end of the administration period on day 28, the rats were sacrificed and blood samples were collected into clean, dry, and labeled Eppendorf tubes containing heparin as an anticoagulant (7.5 I.U). The collected blood samples were centrifuged at 3600× *g* for 15 min at 4 °C to separate serum from plasma. Serum samples were divided into aliquots and kept in deep freezer at −20 °C for the biochemical and hematological assays of the as described previously by Guidelines for Hematoxylin and Eosin Staining, 2001 [91]. The animal organs were removed, weighed, and used for histopathological examination.

#### 4.6.3. Biochemical Analyses

The serum was assayed for alanine aminotransferase (ALT), aspartate aminotransferase (AST), alkaline phosphatase (ALP), acetyl choline esterase (AchE), total proteins, albumin, urea, and creatinine. Hematological analysis using the SYSMEX Hematology Auto Analyzer (Japan) was conducted to estimate the white blood cell (WBC), platelet counts (PLT), hemoglobin concentration (HGB), hematocrit (HCT), mean corpuscular volume (MCV), mean corpuscular hemoglobin (MCH), mean corpuscular hemoglobin concentration (MCHC), and lymphocyte (LYM).The organs of randomly selected three rats from each group were excised, examined grossly, and subsequently fixed in 15% formalin saline then used for histopathological examination. The fixed tissues were processed by dehydration in a series of graded ethanol concentrations, cleared with xylol, and embedded in paraffin blocks. Fourµ thick sections were stained by Hematoxylin–Eosin stain (H & E) for the histopathological analysis under alight microscope (model OLYMPUS CX 41) at 1200× magnification following Osman A. et al., 2019 and Guidelines for Hematoxylin and Eosin Staining, 2001 [90,91]. 

## 5. Conclusions

Food microbiological control is one of the most important approaches for extending its shelf life, ensuring its safety. Based on obtained results, hydrolyzed egg albumin (HEA) effectively inhibits the pathogenic bacteria, where *L. monocytogenes* was the most sensitive organism. Ultra-performance liquid chromatography (UPLC) of the peptic HEA revealed 44 peptides, 17 of them were dipeptides and the other 27 fractions corresponded to bigger peptides (3–9 amino acids). The dipeptides and big peptides represented 26% and 74% of the total hydrolysate, respectively. Mixtures of HEA (MICs) with antibiotics showed more significant antibacterial activity than either of them individually. The enzymatically modified proteins’ ability to act synergistically with antibiotics may be a new approach, overcoming partially the bacterial antibiotic-resistance. HEA has a broad-spectrum antimicrobial activity [6 G (+) and 4 G (−)], enabling it to partially replace the synthetic antibiotic to significant extents. HEA has abroad-spectrum antimicrobial activity (6 G (+)) enabling its recommendation as a food-safe antibacterial treatment since administering it to Wistar albino rats a high ratio of 2500 mg/kg bodyweight for 28 days did not negatively affect liver or kidney functions or trigger any toxicity signs in the animals.

## Figures and Tables

**Figure 1 antibiotics-09-00901-f001:**
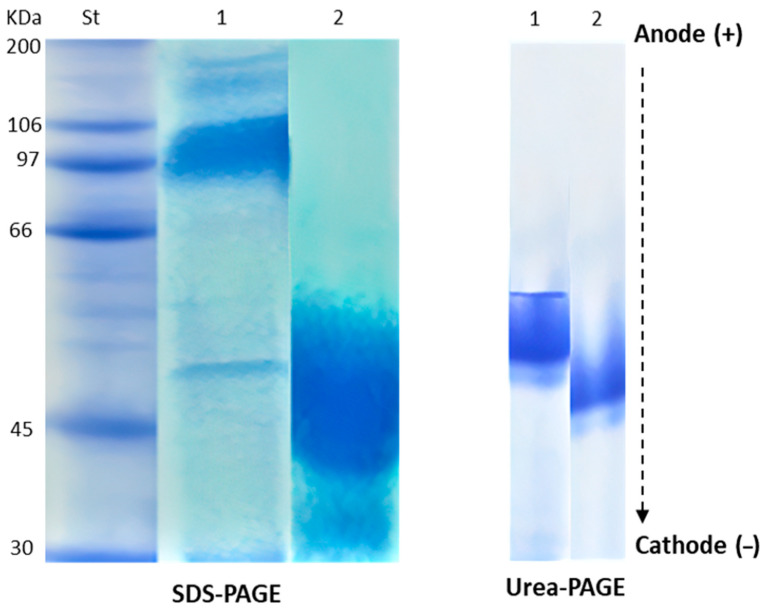
Electrophoretic patterns of native (1) and hydrolyzed (2) egg albumin (NEA and HEA, respectively), either through SDS-PAGE or UREA_PAGE. St is the standard molecular weight proteins. The arrow refers to the direction of protein migration from anode to cathode. The bands in lane (1) representing NEA correspond to 180, 144, 97, 50, and 30 kDa.

**Figure 2 antibiotics-09-00901-f002:**
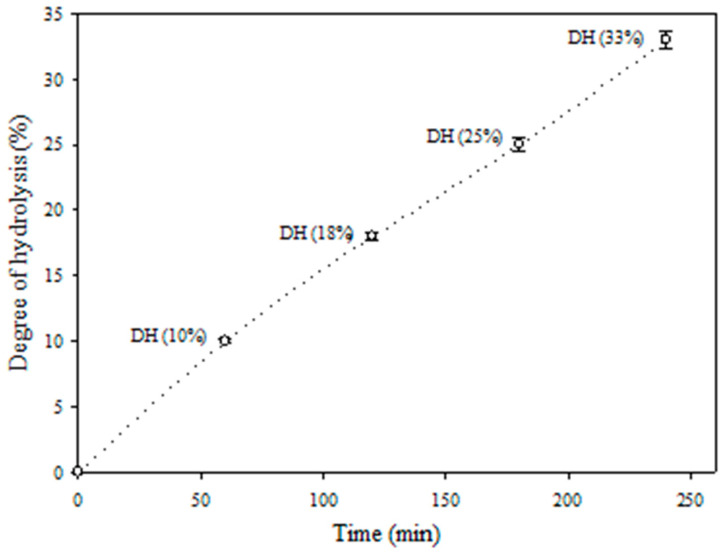
Changes in degree of hydrolysis during enzymatic hydrolysis of egg albumin by pepsin for 240 min.

**Figure 3 antibiotics-09-00901-f003:**
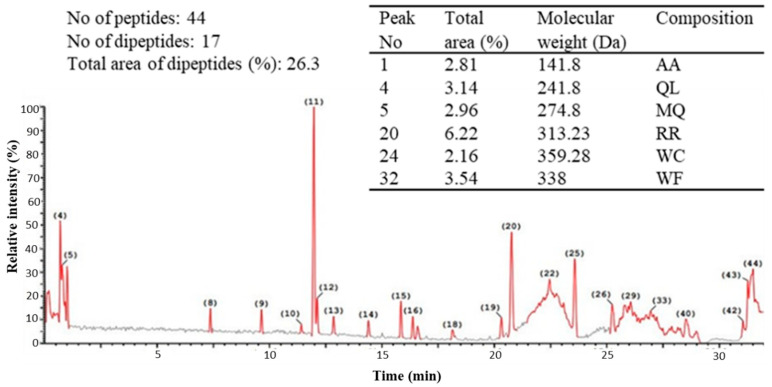
Chromatogram of peptides formation from egg albumin hydrolyzed with pepsin and the mass spectrum of major peaks dipeptides.

**Figure 4 antibiotics-09-00901-f004:**
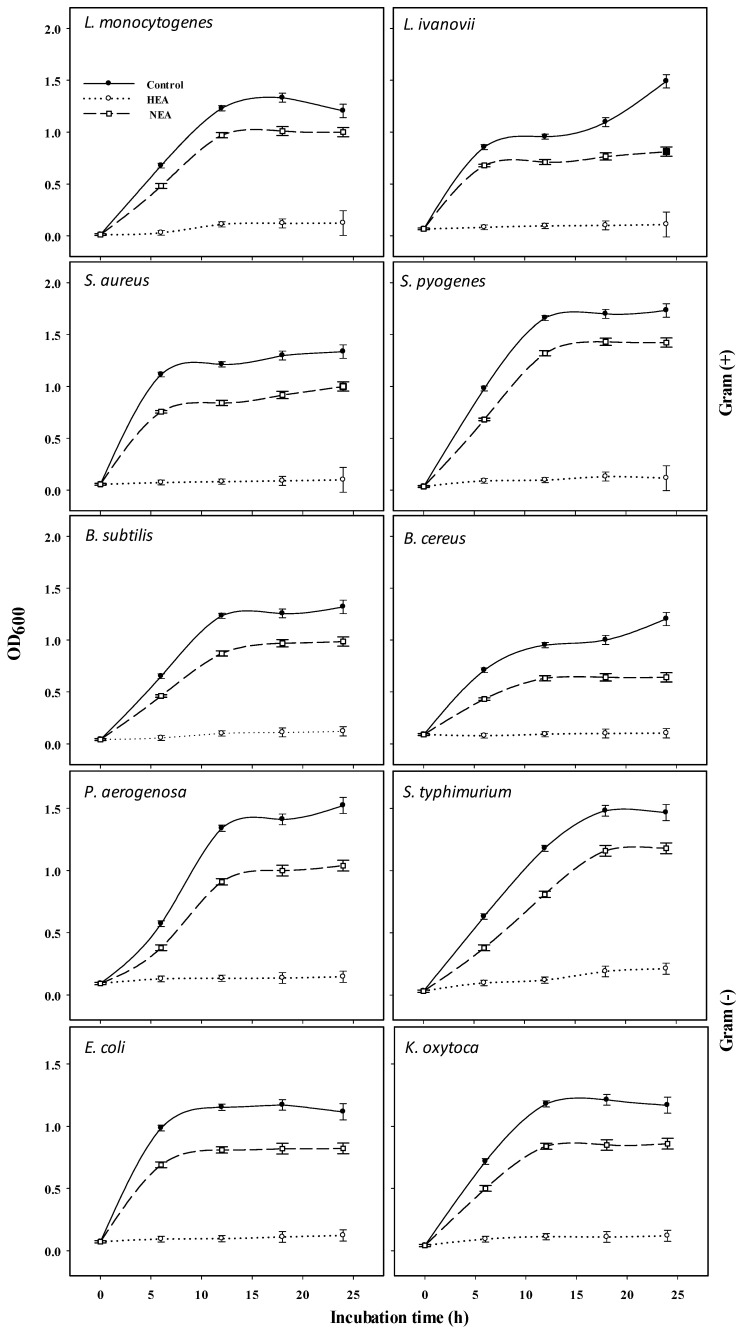
Twenty-four hour growth curves of six pathogenic Gram (+) and four pathogenic Gram (−) in the presence of the value minimum inhibitory concentration (MIC) of HEA.

**Figure 5 antibiotics-09-00901-f005:**
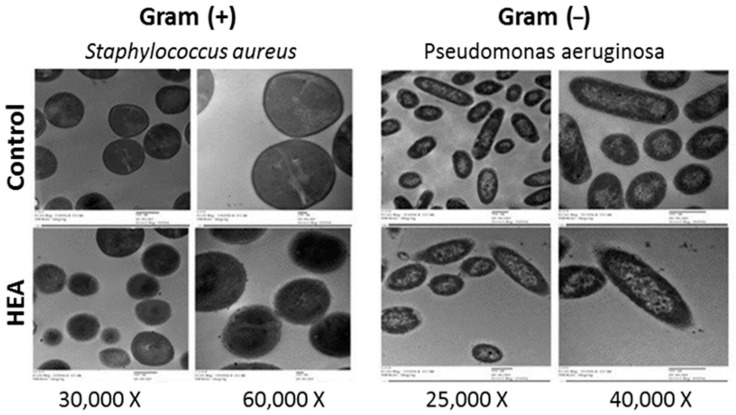
Transmission electron microscopic graphs *Staphylococcus aureus* and *Pseudomonas aeruginosa* as subjected to 100 and 150 μg/mL of hydrolyzed egg albumin, respectively.

**Figure 6 antibiotics-09-00901-f006:**
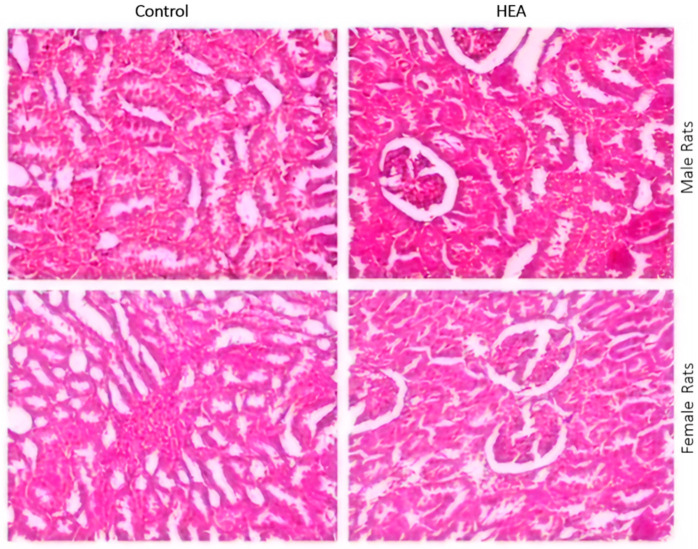
Histopathological examination of H&E stained sections of kidney of male and female albino rats administered 2500 mg/kg body weight of HEA as compared to control (0 mg/kg body weight) using 1200× magnification power.

**Figure 7 antibiotics-09-00901-f007:**
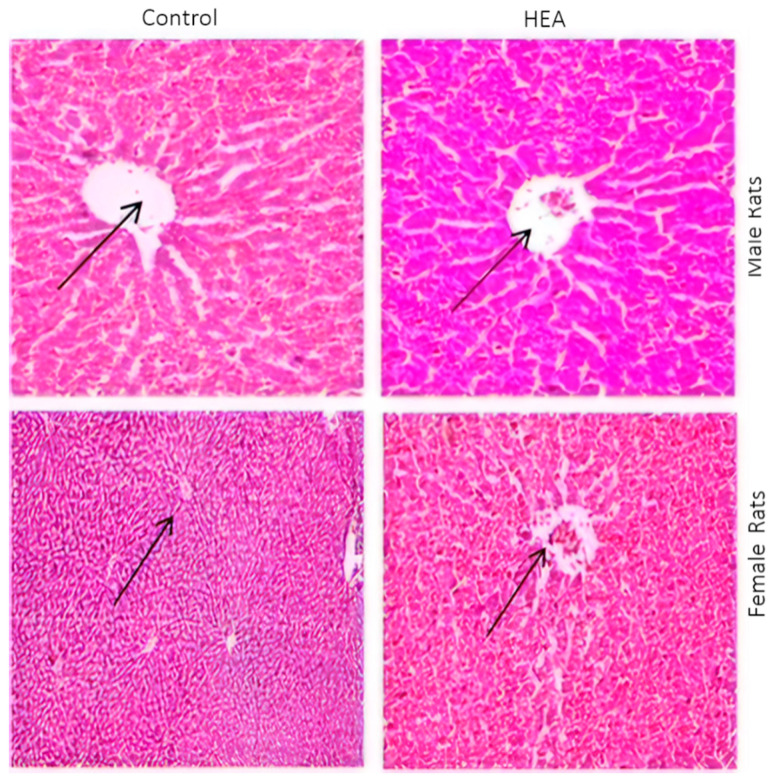
Histopathological examination of H&E stained sections of liver of male and female albino rats administered 2500 mg/kg body weight of HEA as compared to control (0 mg/kg body weight) using 1200× magnification power.

**Table 1 antibiotics-09-00901-t001:** Molecular masses and band intensity of hydrolyzed egg albumin (HEA) compared to native form (NEA).

Band Number	NEA		HEA	
	MW (kDa)	Band Intensity	MW (kDa)	Band Intensity
1	180	1.2	51	10
2	144	1.7	-	-
3	97	6.9	-	-
4	50	1.9	-	-
5	30	2.4	-	-

MW—molecular weight. The numbers 1–5 refer to protein subunits of NEA as shown in the second column. HEA has only one band.

**Table 2 antibiotics-09-00901-t002:** Antibacterial activity of pepsin hydrolyzed egg albumin (HEA) using disc diffusion assay.

	Inhibition Zone Diameter (IZD) (mm/HEA µg mL^−1^)						IZD of NEA (mm/NEA µg mL^−1^)
	0	50	100	150	200	250	250
*L. monocytogens*	0	0	34 ± 1.16 ^a^	36 ± 2.31 ^a^	39 ± 1.73 ^a^	42 ± 2.31 ^a^	7 ± 0.58 ^c^
*K. oxytoca*	0	0	9 ± 0.58 ^d^	28 ± 1.73 ^c^	34 ± 1.73 ^b^	38 ± 1.73 ^abc^	5 ± 0.44 ^d^
*S. pyogenes*	0	0	24 ± 1.73 ^b^	32 ± 1.73 ^b^	35 ± 2.89 ^ab^	39 ± 2.89 ^ab^	3 ± 0.38 ^e^
*L. ivanovii*	0	0	0 ^f^	17 ± 0.58 ^d^	33 ± 1.73 ^b^	34 ± 1.16 ^bc^	10 ± 0.58 ^b^
*S. aureus*	0	0	20 ± 1.16 ^c^	28 ± 1.16 ^c^	30 ± 1.16 ^bc^	33 ± 1.73 ^c^	11 ± 1.05 ^a^
*B. subtilis*	0	0	0 ^f^	9 ± 0.58 ^e^	12 ± 0.58 ^e^	14 ± 0.58 ^e^	12 ± 1.28 ^a^
*P. aerugenosa*	0	0	0 ^f^	9 ± 0.58 ^e^	28 ± 1.73 ^c^	33 ± 1.73 ^c^	3 ± 0.11 ^e^
*S. typhimurium*	0	0	7 ± 0.57 ^de^	13 ± 1.16 ^e^	18 ± 1.16 ^d^	21 ± 1.15 ^d^	1 ± 0.08 ^ef^
*E. coli*	0	0	0 ^f^	0 ^f^	10 ± 0.58 ^e^	13 ± 1.16 ^e^	2 ± 0.09 ^e^
*B. cereus*	0	0	6 ± 0.58 ^e^	11 ± 1.16 ^e^	12 ± 1.16 ^e^	14 ± 1.16 ^e^	2 ± 0.22 ^e^

Every value is the average of three replicates ± SE. Different letters (^a–f^) in the same column refer to significantly different values (*p* < 0.05).

**Table 3 antibiotics-09-00901-t003:** Antibacterial activity of mixed combinations of hydrolyzed egg albumin (HEA) and antibiotic.

Microorganisms	Antibiotic	Antibiotic MIC (µg/mL)	Inhibition Zone (mm)		
			Antibiotic	HEA	Antibiotic: HEA
*L. monocytogenes*	Ciprofloxacin	2	18 ± 1.73 ^c^	34 ± 1.73 ^a^	36 ± 2.31 ^a^
*K. oxytoca*	Gentamycin	6	14 ± 1.16 ^b^	9 ± 0.58 ^c^	16 ± 0.58 ^a^
*S. pyogenes*	Penicillin G	0.016	29 ± 1.16 ^b^	24 ± 1.16 ^c^	34 ± 1.16 ^a^
*L. ivanovii*	Chloramphenicol	7.5	19 ± 1.73 ^b^	17 ± 1.16 ^b^	29 ± 1.16 ^a^
*S. aureus*	Vancomycin	1	14 ± 0.58 ^c^	20 ± 1.16 ^b^	34 ± 0.58 ^a^
*B. subtilis*	Chloramphenicol	2	18 ± 1.16 ^b^	9 ± 1.16 ^c^	20 ± 1.16 ^a^
*P. aeruginosa*	Ciprofloxacin	1	28 ± 1.73 ^b^	9 ± 0.58 ^c^	30 ± 1.16 ^a^
*S. typhimurium*	Ciprofloxacin	4	18 ± 1.16 ^a^	7 ± 0.58 ^b^	19 ± 1.16 ^a^
*E. coli*	Gentamycin	1	19 ± 1.16 ^a^	10 ± 1.16 ^c^	15 ± 1.16 ^b^
*B. cereus*	Vancomycin	4	5 ± 0.29 ^b^	6 ± 0.29 ^b^	15 ± 0.58 ^a^

Every value is the average of three replicates ± SE. Different letters (^a–c^) in the same rowrefer to significantly different values (*p* < 0.05).

**Table 4 antibiotics-09-00901-t004:** Body weight gain (g) of male and female albino rats treated orally with daily hydrolyzed egg albumin (HEA) for 28 days.

Treatment	Control	Tested Proteins Doses (mg/kg b. wt *./day)	
		HEA	
		500	2500
Male			
Initial body wt (g)	165 ± 2.9	163 ± 2.9	160 ± 2.9
Final body wt (g)	245 ± 5.8	249 ± 3.5	250 ± 2.9
Body wt gain (g)	80 ± 2.9	86 ± 2.3	90 ± 5.8
Female			
Initial body wt (g)	135 ± 2.9	140 ± 5.1	135 ± 2.9
Final body wt (g)	187 ± 5.8	196 ± 2.3	196 ± 5.8
Body wt gain (g)	52 ± 1.7 ^b^	56 ± 2.3 ^ab^	61 ± 2.9 ^a^

* b. wt; body weight. Every value is the average of 10 replicates ± SE. Different letters (^a,b^) in the same row refer to significantly different values (*p* < 0.05).

**Table 5 antibiotics-09-00901-t005:** Effect of hydrolyzed egg albumin (HEA) on organ weight (g) of male and female albino rats treated orally daily for 28 days.

Organ Weight (g)	Control	Tested Proteins Doses (mg/kg b. wt./day)	
		HEA	
		500	2500
Male			
Liver	6.3 ± 0.35	6.23 ± 0.14	6.16 ± 0.26
Kidney	1.45 ± 0.13	1.37 ± 0.13	1.28 ± 0.06
Brain	1.37 ± 0.09	1.36 ± 0.14	1.32 ± 0.09
Spleen	0.65 ± 0.07	0.65 ± 0.07	0.63 ± 0.05
Lung	1.27 ± 0.08	1.27 ± 0.08	1.25 ± 0.06
Heart	0.98 ± 0.11	0.94 ± 0.08	0.95 ± 0.07
Testis	1.28 ± 0.13	1.26 ± 0.12	1.24 ± 0.13
Female			
Liver	5.8 ± 0.23	5.78 ± 0.21	5.68 ± 0.15
Kidney	1.29 ± 0.11	1.26 ± 0.08	1.24 ± 0.09
Brain	1.27 ± 0.07	1.26 ± 0.08	1.24 ± 0.15
Spleen	0.61 ± 0.06	0.6 ± 0.04	0.58 ± 0.09
Lung	1.15 ± 0.12	1.16 ± 0.08	1.14 ± 0.06
Heart	0.85 ± 0.05	0.84 ± 0.13	0.83 ± 0.07
Ovary	0.067 ± 0.02	0.064 ± 0.03	0.062 ± 0.02

**Table 6 antibiotics-09-00901-t006:** Effect of hydrolyzed egg albumin (HEA) on hematological parameters of male and female albino rats treated orally daily for 28 days.

Blood Biochemical Parameters	Control	Tested Proteins Doses (mg/kg b. wt./day)	
		HEA	
		500	2500
Male			
WBC ^a^ (×10^3^/µL)	7.12 ± 0.13	7.09 ± 0.07	7.1 ± 0.13
RBC ^b^ (×10^6^ µL)	8.23 ± 0.11	8.23 ± 0.09	8.22 ± 0.12
HGB ^c^ (g/dL)	12.89 ± 0.23	12.89 ± 0.34	12.93 ± 0.27
HCT ^d^ (%)	42.7 ± 0.31	42.12 ± 0.12	42.9 ± 0.17
MCV ^e^ (FL)	51.76 ± 0.17	51.56 ± 0.19	51.65 ± 0.34
MCH ^f^ (pg)	16.98 ± 0.17	16.96 ± 0.13	16.94 ± 0.11
MCHC ^g^ (g/dL)	34.45 ± 0.15	34.44 ± 0.11	34.43 ± 0.09
PLT ^h^ (×10^3^/µL)	820 ± 3.45	827 ± 5.91	822 ± 4.9
LYM ^i^ (%)	5.27 ± 0.17	5.65 ± 0.04	5.16 ± 0.06
Female			
WBC ^a^ (×10^3^/µL)	8.02 ± 0.12	8.01 ± 0.08	8.06 ± 0.12
RBC ^b^ (×10^6^ µL)	8.25 ± 0.11	8.22 ± 0.12	8.21 ± 0.11
HGB ^c^ (g/dL)	12.99 ± 0.21	12.94 ± 0.31	12.96 ± 0.22
HCT ^d^ (%)	44.8 ± 0.31	43.19 ± 0.15	44.9 ± 0.07
MCV ^e^ (FL)	53.70 ± 0.11	52.96 ± 0.14	53.68 ± 0.14
MCH ^f^ (pg)	17.55 ± 0.12	17.86 ± 0.14	17.90 ± 0.21
MCHC ^g^ (g/dL)	36.41 ± 0.16	36.04 ± 0.09	36.33 ± 0.19
PLT ^h^ (×10^3^/µL)	880 ± 62	885 ± 5.44	882 ± 9.51
LYM ^i^ (%)	6.34 ± 0.11	6.25 ± 0.08	6.34 ± 0.09

^a^ White blood cell. ^b^ Red blood cell. ^c^ Hemoglobin concentration. ^d^ Hematocrit (%). ^e^ Mean corpuscular volume. ^f^ Mean corpuscular hemoglobin. ^g^ Mean corpuscular hemoglobin concentration. ^h^ Platelets. ^i^ Lymphocyte.

**Table 7 antibiotics-09-00901-t007:** Effect of hydrolyzed egg albumin (HEA) on blood biochemical parameters of male and female albino rats treated orally daily for 28 days.

Blood Biochemical Parameters	Control	Tested Proteins Doses (mg/kg b. wt./day)	
		HEA	
		500	2500
Male			
ALT (U/mL)	44.82 ± 2.30 ^a^	37.38 ± 1.71 ^b^	34.27 ± 1.15 ^b^
AST (U/mL)	113.56 ± 4.05	103 ± 4.05	98.05 ± 5.84
ALP (U/L)	145.512 ± 5.8	140.05 ± 5.8	134.54 ± 5.81
AChE (U/L)	393.45 ± 5.8	376.02 ± 11.60	371.36 ± 5.80
T.P. (g/dL)	6.43 ± 0.58	6.18 ± 0.58	5.92 ± 0.17
Alb. (g/dL)	3.28 ± 0.12	3.13 ± 0.11	3.08 ± 0.91
Urea (mg/dL)	36.78 ± 3.50	32.08 ± 1.15	31.23 ± 1.15
Creatinine (mg/dL)	0.48 ± 0.045	0.47 ± 0.024	0.42 ± 0.052
Female			
ALT (U/mL)	38.32 ± 2.91	38.02 ± 2.91	36.04 ±2.34
AST (U/mL)	97.19 ± 5.82	96.82 ± 3.51	93.39 ± 2.94
ALP (U/L)	114.26 ± 5.81	113.53 ± 4.11	109.25 ± 5.80
AChE (U/L)	412.65 ± 11.60	408.21 ± 5.80	405.03 ± 3.70
T.P. (g/dL)	6.38 ± 0.58	6.37 ± 0.09	6.33 ± 0.07
Alb. (g/dL)	3.31 ± 0.58	3.29 ± 0.29	3.25 ± 0.14
Urea (mg/dL)	40.32 ± 2.91	39.83 ± 3.51	36.83 ± 3.51
Creatinine (mg/dL)	0.47 ± 0.037	0.47 ± 0.04	0.45 ± 0.067

Every value is the average of ten replicates ± SE. Different letters (^a,b^) in the same row refer to significantly different values (*p* < 0.05).

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
