# Peer review of "Powerful Antibacterial Peptides from Egg Albumin Hydrolysates"

_antibiotics, 2020, doi:10.3390/antibiotics9120901_

Round 1
Reviewer 1 Report
Manuscript: Powerful antibacterial peptides from egg albumin hydrolysates have been evaluated for publication in Antibiotics and publication of the paper is proposed after a comprehensive – major review.
In the present study, native egg albumin was isolated from hen eggs and hydrolyzed with pepsin to produce hydrolyzed egg albumin which was further chemically characterized and screened for its antibacterial activity against 10 pathogenic bacteria.
Comments: The authors are suggested to improve:
Several typing errors are present in the text.
Picture quality - the resolution of the images is poor.
The use of uniform fonts is recommended.
Orderly and uni-form citations of references through the article should be used.
In all tables (2-7) no uniform number of decimal places is used, some data is presented without, some with one and some with two decimal places.
The abstract is poorly written and should be improved, especially the formulations in the second part like HEA may, HEA did not, HEA did not should be improved.
Selection criteria for pathogenic bacteria studied are not mentioned/discussed.
UPLC analysis: The authors claim the presence of 44 peptides - (Figure 3), but the separation in some regions 20-25 min, 25-28 min, 30-32 min is questionable and problematic; in my opinion, several peptides were not identified. Another 5 min »post delay« is mentioned, but during this time 3 peptides were identified ? and it is obvious that the signal of some compounds is still present. An additional optimization of UPLC analysis is therefore necessary.
The references should be presented in a uniform way.
Author Response
|
Replies to reviewer 1# |
|
Manuscript: Powerful antibacterial peptides from egg albumin hydrolysates have been evaluated for publication in Antibiotics and publication of the paper is proposed after a comprehensive – major review. |
|
The authors thanks reviewer 1 for his good job. |
|
In the present study, native egg albumin was isolated from hen eggs and hydrolyzed with pepsin to produce hydrolyzed egg albumin which was further chemically characterized and screened for its antibacterial activity against 10 pathogenic bacteria. |
|
That is true. |
|
Comments: The authors are suggested to improve: Several typing errors are present in the text. |
|
All suggested changes have been responded to. The text was extensively revised for typing errors. |
|
Picture quality - the resolution of the images is poor. |
|
The picture quality in Figure 1 and Figure 7 has been improved. |
|
The use of uniform fonts is recommended. |
|
A uniform font was applied all over the article text. |
|
Orderly and uni-form citations of references through the article should be used. |
|
References were revised so as to be uniform and orderly. |
|
In all tables (2-7) no uniform number of decimal places is used, some data is presented without, some with one and some with two decimal places. |
|
The number of decimals has been unified in the table 2-7. |
|
The abstract is poorly written and should be improved, especially the formulations in the second part like HEA may, HEA did not, HEA did not should be improved. |
|
The abstract was revised and improved accordingly. |
|
Selection criteria for pathogenic bacteria studied are not mentioned/discussed. |
|
This Gram (-) bacteria (especially P. aeruoginosa) is known for being notorious for its ability to survive in the environment, particularly in moist conditions. It may contaminate medicines, surgical equipment, clothing, and dressing with the ability to cause serious infections in immune compromised patients [74].Therefore, it was used in this study for TEM studies.Also, S. aureus bacterium was selected from the Gram (+) group as it was more sensitivity and because of its public health concerns. S. aureusis a pathogen of skin abscesses, pharyngitis, sinusitis, meningitis, pneumonia, endocarditis, osteomyelitis, toxic shock syndrome, sepsis and wound infections following surgery. Moreover, it responsible for food poisoning illness as it is capable of producing several virulence factors such as enterotoxins, adhesins, hemolysins, invasins, superantigens and surface factors that inhibit its phagocytic engulfment [73,75,76]. |
|
UPLC analysis: The authors claim the presence of 44 peptides - (Figure 3), but the separation in some regions 20-25 min, 25-28 min, 30-32 min is questionable and problematic; in my opinion, several peptides were not identified. Another 5 min »post delay« is mentioned, but during this time 3 peptides were identified ? and it is obvious that the signal of some compounds is still present. An additional optimization of UPLC analysis is therefore necessary. |
|
Actually, the UPLC analysis was conducted in an exterior lab with long experience in peptide analysis. The analysis has been confirmed and validated. We tried as much as possible to interpret results in a way to reflect and clarify nature of the resulting peptides. So far, the result showed the multiplicity of the produced peptideswhich are in accordance with the previously known broad specificity of pepsin. So the presence of 44 peaks representing 44 peptides is an expected character of this peptide mixture. The other result indicating that the majority of peptides (27 peptides) corresponded to peptides ranging from tri-peptides to octa- or nona-peptides, forming about 74% of the total peptides. This also agrees with the fact that this hydrolysate was obtained after a limited hydrolysis time (4h). This result led us to conclude that the higher relative quantity of the bigger sized peptides (3-9 amino acids) may indicate their higher participation in the biological activity of the hydrolysates. We estimate that the results obtained from UPLC gives an approximate image on the chemical constitution of the used hydrolysate and aids to link it to its biological activities. We assume this is appropriate for the objective of our current article. However, getting more specific information on the different individual peptides may need more sophisticated analytical tools which are not available for us in the time being. |
|
The references should be presented in a uniform way. |
|
The references were revised to be presented in a uniform way. |

Reviewer 2 Report
Reviewer Comments to Author:
Title: Powerful antibacterial peptides from egg albumin hydrolysates
Journal: antibiotics
The manuscript entitled “Powerful antibacterial peptides from egg albumin hydrolysates” describes the antimicrobial activity and mode of action of protein hydrolysates produced from egg albumin by enzymatic digestion using pepsin. The toxicity has also been investigated. After reviewing the current manuscript I found the concept interesting but the manuscript is prepared too poor. Therefore, the manuscript needs to be revised for many reasons:
- Results and discussion section has to be improved considerably. The author should be focused more on the obtained results and comparing them with others.
- Please check the spaces between the words in whole manuscript. In the pdf version that I received are a lot of mistakes regarding the spaces.
- Please change “Gram +” and “Gram –“ to Gram (+) and Gram (-).
- Please standardize the style fount and of the text.
- Please add the reference for the respective sentences: “The use of synthetic agents is precluded owing to the possible negative impact of such chemicals on the environment and human health.”
- Please standardize the style of the writing the Figures in the whole text. You have to choose one style. I recommend you to use the style Figure 1 not Figure (1) or (Figure 1) when you write that the date are presented in Figure 1.
- The Figures 1, 3, 4, 5, and 6 have to be improved. Low quality resolution.
- Table 1: what does mean the number 1-5? Why on the figure 1 I see the MW 200, 106, 97, 66, 45, 30 but in the table 1 - 180, 144, 97, 50, 30. Could explain why?
- For Figure 2 should the author present a short mechanism ?
- What does mean in the respective sentences “The higher relative quantity of the bigger sized peptides (3-9 amino acids) may indicate their higher participation in the biological activity of the hydrolysates” - biological activity. Could the author specify which one.
- Figure 3: In the figure the author has found 44 signals, in the table are presented only 6 (selected one). Why the author has chosen only 6 and why the respective one.
- Figure 4 it is very difficult to follow. Please improve it.
- Why such concentrations have been chosen for MIC of antibiotics. Please add the information according to which guide you have chosen the respective concentrations.
- Why have been chosen such bacteria for the antimicrobial activity. Please add this information.
- Table 3: if in the materials and methods has been written that for the antibacterial activity analysis of mixed combinations have been used the MIC value why for aureus it has been taken 150 µg/ml not 100 µg/ml. please check it and clarify.
- Fig 5: what does mean G+ and G- please clarify. I suggest to use Gram (+) and Gram (-).
- Please check the sentence: “The faster migration from anode to cathode due to the increasing of positive charges on the hydrolyzed proteins; these protein fractions based on the above data have cationic and hydrophobic nature [63, 64].”
- Materials and method section: please change the “…disc-diffusion method as described previously by Schafer E.W. et al. 1972 [88].” Please check and improve in whole text this style of writing.
Author Response
|
Replies to reviewer 2# |
|
The manuscript entitled “Powerful antibacterial peptides from egg albumin hydrolysates” describes the antimicrobial activity and mode of action of protein hydrolysates produced from egg albumin by enzymatic digestion using pepsin. The toxicity has also been investigated. After reviewing the current manuscript I found the concept interesting but the manuscript is prepared too poor. Therefore, the manuscript needs to be revised for many reasons: |
|
Thanks for Reviewer 2 for appreciating the article concept and for his recommendations to improve the article. |
|
1- Results and discussion section has to be improved considerably. The author should be focused more on the obtained results and comparing them with others. |
|
Results and discussion have been revised and improved so as to be more focused on the current results.. |
|
2- Please check the spaces between the words in whole manuscript. In the pdf version that I received are a lot of mistakes regarding the spaces. |
|
Thanks for Reviewer 2 for this point. We checked the spaces between the words in whole manuscript. |
|
3- Please change “Gram +” and “Gram –“ to Gram (+) and Gram (-). |
|
Thank you very much. The “Gram +” and “Gram –“ changed to Gram (+) and Gram (-)in whole manuscript.. |
|
4- Please standardize the style fount and of the text. |
|
Thank you. It was done. |
|
5- Please add the reference for the respective sentences: “The use of synthetic agents is precluded owing to the possible negative impact of such chemicals on the environment and human health.” |
|
We added the reference (Akyil, D., Ozkara, A., & Konuk, M. (2016). Pesticides, Environmental Pollution, and Health. Chapters. http://dx.doi.org/10.5772/63094 |
|
6- Please standardize the style of the writing the Figures in the whole text. You have to choose one style. I recommend you to use the style Figure 1 not Figure (1) or (Figure 1) when you write that the date are presented in Figure 1. |
|
The style of the writing the Figures in the whole text was standardized. |
|
7- The Figures 1, 3, 4, 5, and 6 have to be improved. Low quality resolution. |
|
The resolution of all figures has been improved. |
|
8- Table 1: what does mean the number 1-5? Why on the figure 1 I see the MW 200, 106, 97, 66, 45, 30 but in the table 1 - 180, 144, 97, 50, 30. Could explain why? |
|
The numbers 1-5 refer to the protein fractions of our tested protein (egg albumin) from up to down successively. On Figure 1, the MW 200, 106, 97, 66, 45, 30 refer to molecular weight standards in the first lane (St) in Figure 1. In Table 1 the MW 180, 144, 97, 50, 30.refer to the subunits of egg albumin as also shown in the second lane of figure 1 (1). These molecular weights were deduced using the known standard MW in lane. |
|
9- For Figure 2 should the author present a short mechanism ? We added small statement to the third papragrah of discussion (marked in yellow).
{The obtained hydrolysate style, showing increasingly and steadily rate of hydrolysis with time during 4 h, reflects the nature of pepsin specificity which tends to attacks protein molecules gradually from the C- and N-terminals.[54,55]. |
|
10- What does mean in the respective sentences “The higher relative quantity of the bigger sized peptides (3-9 amino acids) may indicate their higher participation in the biological activity of the hydrolysates” - biological activity. Could the author specify which one. |
|
The sentence “The higher relative quantity of the bigger sized peptides (3-9 amino acids) may indicate their higher participation in the biological activity of the hydrolysates” means that peptide more than 3 amino acids may be essentially more responsible for the bioactivity since they represent more than 74% of the peptide mixture. However, the data available from the analysis can not give determine which peptide was more responsible. |
|
11- Figure 3: In the figure the author has found 44 signals, in the table are presented only 6 (selected one). Why the author has chosen only 6 and why the respective one. |
|
In the table we represented only the major ones which are expected to give the highest biological impact. |
|
12- Figure 4 it is very difficult to follow. Please improve it.
|
|
Figure 4 was considerably improved.
|
|
13- Why such concentrations have been chosen for MIC of antibiotics. Please add the information according to which guide you have chosen the respective concentrations. |
|
MIC of HEA and five antibiotics; ciprofloxacin (10 μg), gentamycin (10 μg), penicillin G (10 μg), chloramphenicol (30 μg) and vancomycin (30 μg) was evaluated using standard inoculums of about 2×105 CFU/ mL as described previously[35].
Results of antibiotic sensitivity were taken according to the instructions reported previously[89, 90] |
|
14- Why have been chosen such bacteria for the antimicrobial activity. Please add this information.
|
|
This Gram (-) bacteria (especially P. aeruoginosa) is known for being notorious for its ability to survive in the environment, particularly in moist conditions. It may contaminate medicines, surgical equipment, clothing, and dressing with the ability to cause serious infections in immune compromised patients [74].Therefore, it was used in this study for TEM studies.Also, S. aureus bacterium was selected from the Gram (+) group as it was more sensitivity and because of its public health concerns. S. aureusis a pathogen of skin abscesses, pharyngitis, sinusitis, meningitis, pneumonia, endocarditis, osteomyelitis, toxic shock syndrome, sepsis and wound infections following surgery. Moreover, it responsible for food poisoning illness as it is capable of producing several virulence factors such as enterotoxins, adhesins, hemolysins, invasins, superantigens and surface factors that inhibit its phagocytic engulfment [73,75,76].
|
|
15- Table 3: if in the materials and methods has been written that for the antibacterial activity analysis of mixed combinations have been used the MIC value why for aureus it has been taken 150 µg/ml not 100 µg/ml. please check it and clarify.
|
|
We took 100 µg/ml in all experiments.
A number of test tubes, containing 10 mL of BHIB (Oxoid, United Kingdom); wereinoculated with 100 µL of bacterial suspension (2×105 CFU/ mL ) at the log phase and combined with100 μg/mL HEA for L. monocytogenes, B. cerueus, S. aureus,S. typhimurium, S. pyogenesand K. oxytocaand 150 μg/mL HEA for P. aeruginosa, B. subtilis and L. ivanovii and 200 μg/mL HEA for E. coli. The presence of HEA (100 μg/mL) in BHI broth media containing S. aureus (Gram (+)
|
|
16- Fig 5: what does mean G+ and G- please clarify. I suggest to use Gram (+) and Gram (-).
|
|
We corrected the information on Figure 5. Accordingly.
|
|
17- Please check the sentence: “The faster migration from anode to cathode due to the increasing of positive charges on the hydrolyzed proteins; these protein fractions based on the above data have cationic and hydrophobic nature [63, 64].”
|
|
This sentence was modified into a clearer form: [The faster migration from anode to cathode, in case of the hydrolyzed protein than the native one refers to bigger positive charges. This may imply the liberation of some cationic peptides in the course of hydrolysis in accordance with [63, 64].
|
|
18- Materials and method section: please change the “…disc-diffusion method as described previously by Schafer E.W. et al. 1972 [88].” Please check and improve in whole text this style of writing.
|
|
It was corrected. disc-diffusion method as described previously [88]. |

Round 2
Reviewer 1 Report
The manuscript "Powerful antibacterial peptides from egg albuminhydrolysates" was again evaluated for publication in Antibiotics.
The authors have improved the article and additionally commented
on the results, therefore the publication of the paper
in its present form is suggested.
Author Response
|
Reviewer 1 |
|
The manuscript "Powerful antibacterial peptides from egg albumin |
|
We thank reviewer 1 very much. We are happy to respond to reviewer 1 |

Reviewer 2 Report
The manuscript has not be improved at all. Some comments the authors have not taken in consideration. Therefore, please find few more comments:
- Please check the spaces between the words in whole manuscript (especially in the new added information).
- Please standardize the style fount and of the text (especially in the new added information).
- The resolution of the figures have not be improved. The Figures 1, 3, 4, 5, and 6 have to be improved. Please be focused on the resolution of the picture and on the fond of the figures. Even the figure 4 that has been improved is not acceptable, due to the low resolution of the figure. Please also check the dpi of the figures according to the journal quid. In such style the manuscript cannot be accepted for publication.
- Where are specified the bands 1-5 in the figure. In the table is clear but in the figure no. Once you make correlation between the figure and table, should be specified in the figure, as well. This correlation is not see from the table and figure and is difficult for the reader to follow the idea.. Please provide it.
- Has been this explanation added to the manuscript to allowed the reader to follow the idea? “The numbers 1-5 refer to the protein fractions of our tested protein (egg albumin) from up to down successively. On Figure 1, the MW 200, 106, 97, 66, 45, 30 refer to molecular weight standards in the first lane (St) in Figure 1. In Table 1 the MW 180, 144, 97, 50, 30.refer to the subunits of egg albumin as also shown in the second lane of figure 1 (1). These molecular weights were deduced using the known standard MW in lane.”
- Has been this information added to the manuscript for the reader? “The sentence “The higher relative quantity of the bigger sized peptides (3-9 amino acids) may indicate their higher participation in the biological activity of the hydrolysates” means that peptide more than 3 amino acids may be essentially more responsible for the bioactivity since they represent more than 74% of the peptide mixture. However, the data available from the analysis can not give determine which peptide was more responsible”. Moreover, which biological activity, specified exactly and add to the text, please.
- Has been added this explanation to the manuscript, for the reader? “In the table we represented only the major ones which are expected to give the highest biological impact”.
- If the author used 100 µg/ml in all experiments why the data from the table 2 and 3 for Aureus are not correlated. According to the table 2 the MIC is 20±1.16 but in table 3, 28±1.16. Please explain.
- The author did not take in consideration the Reviewer’s remark regarding the writing of the idea concerning the “previous work”. In Materials and method section the author has such sentence ““…disc-diffusion method as described previously [88]”, the sentences should be written in such style: “…disc-diffusion method as described previously by Schafer E.W. et al. 1972 [88].” Please check and improve in whole manuscript this style of writing.

Author Response
|
Reviewer 2 |
|
The manuscript has not be improved at all. Some comments the authors have not taken in consideration. Therefore, please find few more comments: |
|
We improved the manuscript as much as possible and responded to all reviewers’ comments. However, we thank reviewer 2 for discovering more weak points to correct. We are happy to respond to each point in the following lines. |
|
Comments and Suggestions for Authors |
|
|
The spaces were double checked all over the text. |
|
|
The font style was standardized to the font Palatino Linotype, recommended by MDPI. |
|
|
Actually, we highly improved the images in Figures 1, 6 and 7 and reconstituted Figure 4. We have checked again and enhanced further the resolutions of all figures. Thanks to the reviewer’s advices, the figures look much better now. |
|
|
We specified the bands 1-5 in Figure 1 as follows (The bands in lane (1) representing NEA correspond to 180, 144, 97, 50 and 30 kDa). |
|
|
This explanation was also added to the manuscript under section 2.1, in the first paragraph. |
|
|
This information was added in the discussion section (paragraph 4). |
|
|
This explanation was added at the end of section 2.2. |
|
|
Yes, the reviewer is right. Actually, this a mistake in data transfer from Table 2 to Table 3. We corrected the transferred value in Table 3 to be 20±1.16. All other data are consistent between tables 2 and 3. This a very careful and excellent reviewing.Thanks again for the reviewer. |
|
|
All references were corrected accordingly in Materials and Methods. |

Round 3
Reviewer 2 Report
- Please change the sentence “In the Table included in Figure 3, the major constituting peptides are only presented since they expected to produce the highestbiological impact” to the “The selected peptides summarised in Table included in Figure 3, have been chosen as the major constituting peptides that are expected to produce the highest biological impact”
- Please change “....as described previously Guidelines for Hematoxylin and Eosin Staining, 2001[91].” To the as described previously by Guidelines for Hematoxylin and Eosin Staining, 2001[91].
- Please add the space in this sentence “...The presence of HEA (100μg/mL) in BHI broth media containing S. Aureus (Gram (+)bacterium)...”
- This Figure has to be improve. The numbers from the axe x are not fully seen.
- Please improve the quality of the Figure (marked places)

Author Response
|
Reviewer 2 |
|
Comments and Suggestions for Authors |
|
|
The sentence was changed in the text. |
|
|
It was changed. |
|
|
The space was added. |
|
|
The Figure improvement done. |
|
|
The Figure quality was improved.
This very careful and excellent reviewing. Thanks for the reviewer for additional remarks and recommendations. |
